# Sequential Automated Machine Learning: Bandits-driven Exploration using a Collaborative Filtering Representation

**Maxime Heuillet**
*Institute Intelligence and Data, Laval University*

**Benoit Debaque**
*Thales Research and Technology*

**Audrey Durand**
*Canada CIFAR AI Chair, Institute Intelligence and Data*

## Abstract

The goal of Automated Machine Learning (AutoML) is to make Machine Learning (ML) tools more accessible. Collaborative Filtering (CF) methods have shown great success in automating the creation of machine learning pipelines. In this work, we frame the AutoML problem under a sequential setting where datasets arrive one at a time. On each dataset, an agent can try a small number of pipelines (exploration) before recommending a pipeline for this dataset (recommendation). The goal is to maximize the performance of the recommended pipelines over the sequence of datasets. More specifically, we focus on the exploration policy used for selecting the pipelines to explore before making the recommendation. We propose an approach based on the LinUCB bandit algorithm that leverages the latent representations extracted from matrix factorization (MF). We show that the exploration policy impacts the recommendation performance and that MF-based latent representations are more useful for exploration than for recommendation.

## 1. Introduction

The goal of Automated Machine Learning (AutoML) is to make Machine Learning (ML) tools more accessible. The AutoML community has developed frameworks that automate the creation of machine learning pipelines (Feurer et al., 2020; Kotthoff et al., 2017; Olson et al., 2016). Collaborative Filtering (CF) frameworks have also shown great success to tackle this problem (Fusi et al., 2018; Yang et al., 2019, 2020; Zhang et al., 2020; Cunha et al., 2018). Existing work focuses on an off-line setting that requires the generation a large pipeline performances benchmarking used as the training matrix. When a user asks for a pipeline recommendation for a new dataset, an agent selects a small number of pipelines to try on this dataset, then CF is used to infer the performance of unobserved pipelines on this dataset given existing observations from the training matrix. Then, a recommendation is performed based on these predictions.

In this work, we tackle the problem of pipeline optimization in a sequential setting where datasets arrive one after the other. For each dataset, the agent can try a small number of pipelines (exploration policy) to conduct the predictive task on this dataset. The performance observed for these pipelines on the dataset are then added to the knowledge of the agent, it then recommends a pipeline (recommendation policy) for this dataset. The goal of the agent is to recommend a promising pipeline for each dataset encountered in the

sequence. We therefore focus on how the sequential information collected by the exploration policy influences the recommendation performance.

More specifically, we formulate this sequential AutoML problem under the bandit setting (Lattimore and Szepesvári, 2020) and propose a LinUCB (Li et al., 2010) exploration policy by leveraging a CF-based latent representation extracted from matrix factorization. Our empirical results highlight the impact of the exploration policy on recommendations and indicate that recommendations should not be based on inference.

## 2. Problem setting

Consider that there are $K$ pipelines available for conducting a predictive task given a dataset, i.e. to predict the output given the input. Assuming that trying every pipeline would require too much time to identify the most effective one (or computing resources), we would like to identify the optimal pipeline for a new dataset by trying only $c < K$ pipelines on the dataset (where $c$ is small compared to $K$). Given a sequence of datasets, the idea in this work is to leverage information about pipeline performances that was acquired on previously encountered datasets in order to be able to identify the optimal pipeline in a few trials on a new dataset.

This AutoML problem can be formulated as an episodic game, where for each episode $t \in \{1, \ldots, T\}$, a new dataset $d_t$ arrives, then the agent selects a subset $\mathcal{C}(t)$ of $c$ pipelines (exploration policy) and observes the performance $r_{t,k}$ of each pipeline $k \in \mathcal{C}(t)$ applied on $d_t$. These performances are then appended to the knowledge matrix $R(t-1) \in [0,1]^{t-1 \times K}$. More precisely, the resulting knowledge matrix $R'(t) \in [0,1]^{t \times K}$ is a sparse matrix in which the $i$-th row contains the performances that were observed for pipelines in $\mathcal{C}(i)$ on dataset $d_i$ (with $i \leq t$). Note that $R(0)$ is empty (it contains 0 rows). Using $R'(t)$, the agent then recommends what they think is the optimal pipeline, i.e. $k_t$ (recommendation policy). The performance $r_{t,k_t}$ observed for the recommended pipeline on dataset $d_t$ is then added to $R'(t)$ (row $t$, column $k_t$), resulting into the knowledge matrix $R(t)$. Note that if $k_t \in \mathcal{C}(t)$, then $R(t) = R'(t)$. Let $k_t^\star = \arg\max_{k=1\ldots K} r_{t,k}$ denote the optimal pipeline for dataset $d_t$, i.e. the one maximizing performance on dataset $d_t$. The goal of the agent is to explore pipelines $\mathcal{C}(t)$ such as to recommend $k_t$ to minimize the cumulative regret over time:

$$\mathfrak{R}(T) = \sum_{t=1}^{T} (r_{t,k_t^\star} - r_{t,k_t}). \tag{1}$$

This problem corresponds to a bandit problem (Lattimore and Szepesvári, 2020) where $c$ actions can be tried (without regret) before recommending the action on a given time step. In other words, the bandit problem is the specific case with $c = 0$.

## 3. Related work

In the sequential AutoML problem initially introduced by Hutter et al. (2011), pipelines were limited to hyper-parameters of pre-determined models and the task was to optimize the hyper-parameters on a given dataset by sequentially trying different values for that dataset. Lindauer and Hutter (2017) later showed that information from optimization tasks conducted on previously encountered datasets could be used to warm-start the search,

yielding to substantial speed-ups. Similarly, our work tackles the general AutoML problem on a sequence of tasks. However, we are not limited to hyper-parameter tuning, but rather consider the design of ML pipelines that are the combination of a data preparation operation, a classification algorithm and a set of hyper-parameters.

Collaborative Filtering (CF) approaches have been previously applied to the off-line AutoML problem (Fusi et al., 2018; Yang et al., 2020, 2019) where the goal is to recommend a pipeline for a dataset given a provided knowledge matrix. This almost dense matrix typically consists of a benchmarking of many of pipeline performances on several datasets. The focus is therefore to use this knowledge to perform good recommendations for the targeted dataset. In this work, we frame the AutoML problem under an on-line setting where knowledge is gathered and accumulated along a sequence of optimization tasks. The (sparse) knowledge matrix available when encountering a given dataset therefore depends on pipelines explored and recommended on previously encountered datasets.

Recall that the exploration policy is in charge of selecting the pipelines to explore before recommending a pipeline for a given dataset. In the off-line setting, the unique goal of the exploration policy is to select pipelines informative about the current dataset only (Fusi et al., 2018; Yang et al., 2019, 2020). This selection typically relies on a provided knowledge and on meta-information, e.g. a set of meta-features (Fusi et al., 2018) or training time (Yang et al., 2019, 2020). Under the sequential setting tackled in this work, the exploration policy must also select pipelines that allow to share knowledge across datasets encountered over time. A similar challenge occurs in the sequential recommendation system setting where one must recommend items (akin to pipelines) to a new user (akin to a new dataset). Bandit algorithms were shown to be successful at exploring items for a new user in order to simultaneously improve CF-based recommendations while collecting informative knowledge about the users (Guillou et al., 2015; Mary et al., 2015). This inspired the approach proposed in this work.

## 4. A bandit strategy for sequential AutoML

We now introduce a strategy based on the linear bandits (Lattimore and Szepesvári, 2020) algorithm LinUCB (Li et al., 2010) in order to conduct the exploration of pipelines by leveraging a Collaborative Filtering (CF) latent representation of available pipelines.

### 4.1 Learning a latent representation from matrix factorization

CF strategies based on matrix factorization have shown great success as recommendation policies in AutoML problems (Fusi et al., 2018; Yang et al., 2020). These algorithms extract a latent representation for pipelines by decomposing a (possibly sparse) knowledge matrix $R$ acquired on $J$ datasets (rows) and $K$ pipelines (columns) into matrices $P \in \mathbb{R}^{K \times L}$ and $Q \in \mathbb{R}^{L \times J}$ such that $R \approx \langle P, Q \rangle$. The operation $\langle, \rangle$ denotes the scalar product between two matrices and $L$ denotes the size of the latent space ($L < K$ and $L < J$). This is achieved by minimizing an objective function based on the Mean Squared Error (Koren et al., 2009).

A latent representation $(P, Q)$ can be used by the recommendation policy. For example, one can factorize $R'(t)$ into $P'(t)$ and $Q'(t)$, and use the dense reconstruction $\bar{R}'(t) = \langle P'(t), Q'(t) \rangle$ to recommend $k_t = \arg\max_{k=1...K} \bar{r}_{t,k}$, where $\bar{r}_{t,k}$ is the predicted performance of pipeline $k$ on dataset $d_t$.

## 4.2 LinUCB exploration policy

In the sequential AutoML problem, a good exploration policy trades-off the exploration of pipelines to improve the knowledge and the exploitation of pipelines to support the recommendation policy. We propose an exploration policy based on the contextual Upper Confidence Bound (UCB) algorithm with disjoint linear models (Li et al., 2010), using the latent representation extracted from matrix factorization as context.

Given a set of available actions to chose from, the LinUCB algorithm uses a context $\phi_k$ for each available action $k \in \{1, \ldots, K\}$ and selects the action which maximizes an optimistic predicted value using upper confidence intervals. The context corresponds to features allowing to share information across actions. Here actions are pipelines and we consider as context for pipeline $k$ at episode $t$ the $k$-th column of the latent representation $P(t-1)$ factorized from the previous knowledge $R(t-1)$. LinUCB assumes that the problem setting is characterized by an unknown parameter vector $\theta_k^\star$ for each $k$, such that the expected outcome for a pipeline corresponds to $\mathbb{E}[r_{t,k}] = \langle \phi_k, \theta_k^\star \rangle$. At episode $t$ and for each pipeline $k$, the unknown parameter $\theta_k^\star$ is estimated using ridge regression, such that $\hat{\theta}_k(t) = A(t)_k^{-1} b(t)_k$ where $A(t) = A(t-1) + \phi_k \phi_k^\top$ and $b(t)_k = b(t-1)_k + r_{t,k}\phi_k$. Note that $b(0)_k = 0_{K,1}$ and $A(0) = I_K$. Using this, LinUCB computes for each pipeline $k$: i) the estimated reward $\hat{r}_{t,k} = \langle \phi_k, \hat{\theta}_k(t) \rangle$ and ii) the uncertainty of $r_{t,k}$ defined as $u_{t,k} = \sqrt{\phi_k^\top A(t)_k^{-1} \phi_k}$. For a given dataset $d_t$, LinUCB selects as $\mathcal{C}(t)$ the set of top-$c$ pipelines according to their UCB value: $\mathrm{UCB}_k(t) = \hat{r}_{t,k} + \alpha u_{t,k}$, where $\alpha \geq 0$ articulates the exploration-exploitation trade-off.

In practice, the latent representation $P(t)$ of the knowledge $R(t)$ is a highly non-stationary context due to the matrix $R(t)$ being updated at each episode $t$. Unfortunately, LinUCB assumes that the context space is stationary. In order to preserve an illusion of stationarity, we therefore use for the exploration policy a buffered replicate of $P(t)$, denoted $\tilde{P}(t)$, that is updated after every $s$ episodes. We therefore wait $s$ steps before the first update, which corresponds to a burn-in phase during which pipelines in $\mathcal{C}(t)$ are selected uniformly at random. This reduces instabilities in the latent representation. The resulting procedure is detailed in the Algorithm 1 of Appendix A.

## 5. Experiment

We now conduct experiments in order to investigate the impact of the exploration policy on the pipeline recommendation performance using different Collaborative Filtering (CF) algorithms for learning latent representations in the sequential AutoML problem[1].

## 5.1 Methods

We consider two CF-based algorithms for learning latent representations: a bias aware Matrix Factorization (MF-bias, see Koren et al. (2009)) and a non-linear Matrix Factorization (NeuralCF, see He et al. (2017)).

We consider three exploration policies for the selection of $\mathcal{C}(t)$: (i) Random: $c$ pipelines are sampled uniformly (without replacement) among the $K$ pipelines on episode $t$; (ii) Lin-

---

1. The code is available on Github: `https://github.com/MaxHeuillet/sequentialAutoML`.

UCB: see Section 4; (iii) KNN: the agent benefits from the knowledge of an exhaustive benchmark of all available pipelines evaluated on additional 140 datasets, allowing to identify the best pipeline for each of these datasets. A set of normalized meta-features (see Appendix C) is computed on each dataset. For a new dataset $d_t$, the KNN exploration policy computes the set of meta-features on $d_t$. The $c$ best (unique) pipelines are then selected based on the nearest-neighbor order (L1-distance on meta-features). This is a brute brute-force version of the exploration policy from Feurer et al. (2015, 2020); Fusi et al. (2018) which represents the current standard in the literature.

We consider two recommendation strategies for selecting $k_t$: (i) Best of $\mathcal{C}(t)$: recommend the pipeline with the best observed performance from the set of selected pipelines $C(t)$, i.e $k_t = \arg\max_{k \in \mathcal{C}(t)} r_{t,k}$. (ii) Best given CF inference: compute a latent representation $P'(t)$ and $Q'(t)$ from $R'(t)$, and infer a dense knowledge matrix $\bar{R}'(t) = \langle P'(t), Q'(t) \rangle$, and recommend $k_t = \arg\max_{k \in \{1,...,K\}} \bar{r}_{t,k}$, where $\bar{r}_{t,k}$ is the predicted performance of pipeline $k$ on dataset $d_t$.

## 5.2 Evaluation

We articulated a sequential AutoML problem using 666 datasets obtained from the UCR uni-dimensional time series repository (Dau et al., 2019). We considered 175 pipelines, where each pipeline was a combination of a data preparation technique (among 4) with one predictive model (among 7) and different hyper-parameters. The selected pipelines and datasets are described in Appendix B. Hyper-parameters are given in Appendix C. We consider exploration budgets $c \in \{2, 6\}$, which correspond to an exploration order around 1% of the available pipelines ($K = 175$).

Each strategy (exploration policy combined with a recommendation policy) was evaluated by computing its cumulative regret (Equation 1) on a 10-folds cross-validation, where each fold corresponds to a sequence of 526 datasets (sampled from the 666), the remaining subset of 140 datasets being used by the KNN exploration policy. Note that the sequence ordering for a given fold is the same for all strategies in the same fold.

## 5.3 Results

Figure 1 shows the cumulative regret averaged over the 10 folds for every combinations of exploration and recommendation policies, for $s = 10$ steps between matrix factorization updates. Rows correspond to the exploration budgets $c$ and columns correspond to exploration policies.

We first observe that the performance of recommendation policies based on CF inference (plain lines) is highly influenced by the exploration policy. Although the random exploration policy provides a uniform representation of the decision space, the matrix factorization fails to recommend the most efficient pipelines, leading to the worst performance. Best performances with these recommendation policies are always obtained using the LinUCB exploration policy.

We also observe that the best performance is achieved when exploring with the KNN policy and recommending the observed pipeline in $C(t)$ with the highest performance (second column, black dotted line). However, the KNN approach benefits from the additional knowledge of an exhaustive benchmark of the 175 available pipelines over 140 datasets. This

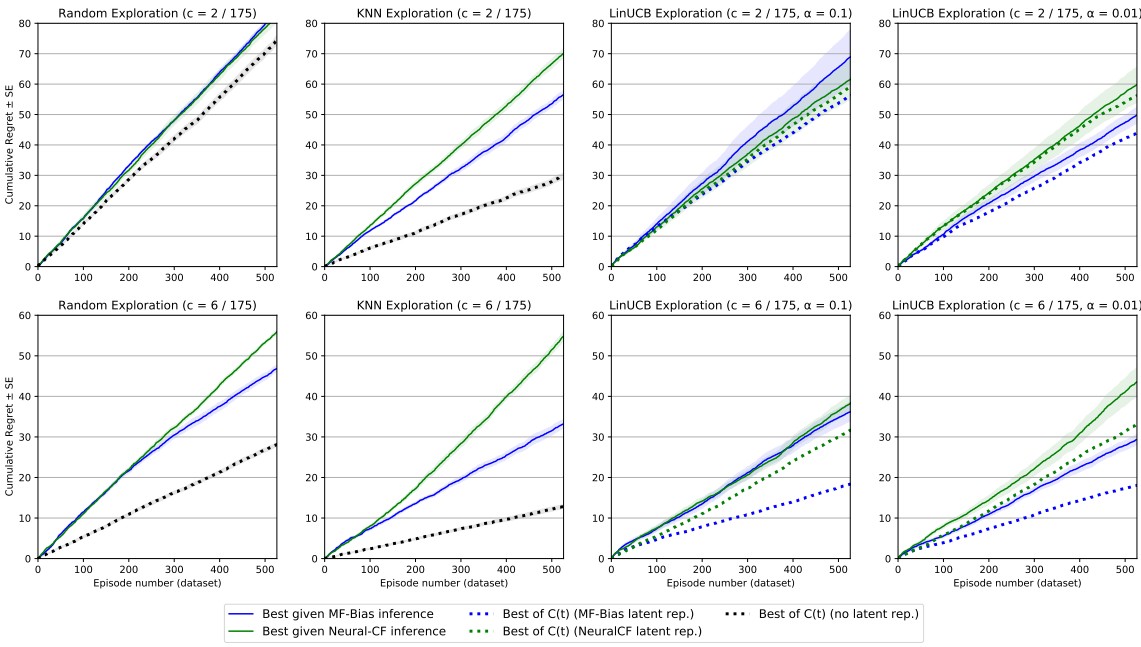

Figure 1: Cumulative regret averaged over a 10-folds cross validation with the resulting standard error using $s = 10$ steps between matrix factorization updates. Lower is better.

corresponds to much more knowledge and computing resources compared with the 2nd best approach, which corresponds to exploring with LinUCB and recommending the observed pipeline in $C(t)$ with the highest performance (columns 3-4, blue dotted line). More importantly we notice that the gap between the KNN-based and the LinUCB-based strategies narrows as $c$ is increased (although still low). This indicates that latent representations are efficient for guiding exploration. This is impressive considering that the KNN-based approach uses a dense knowledge matrix of $140 \times 175 = 24.5k$ observations, while the LinUCB-based approach uses a sparse knowledge matrix of at most $c \times t$ observations for making a decision at time $t$.

Finally, the highest performances being always achieved by recommending the best pipelines over $\mathcal{C}(t)$ suggests that inference may bring additional noise hurting the recommendation. Recommending directly from $C(t)$ is subject to the noise in the $C(t)$ selection, but then is based on the actual performances.

## 6. Conclusion

We introduced the sequential AutoML problem, where the goal is to recommend efficient ML pipelines on a succession of datasets. Our approach based on the LinUCB bandit algorithm leverages the latent representations of matrix factorization (MF) algorithms. Our results highlight the impact of the exploration policy on recommendations and indicate that recommendations should not be based on inference.

## 7. Acknowledgements

This work was funded by a MITACS Accelerate Research grant (IT17584) and it involved the financial support of Thales Research and Technology (Montreal, Canada).

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

## Appendix A. LinUCB exploration policy for sequential AutoML

Algorithm 1 describes in details the procedure. Lines 4-6 indicate the burn-in phase of the first $s$ episodes where the $c$ pipelines in $C(t)$ are selected at random. After the burn-in phase, LinUCB explores pipelines with the highest UCB (Lines 8-12). The performance observed with pipelines in $C(t)$ and recommended pipeline $k_t$ is added to the knowledge (Lines 14-17 and 19-24). Then, it is used for updating ridge regression (Lines 31-34). The latent representation is updated every $s$ episodes (Lines 25-30).

## Appendix B. Experiments benchmark

### B.1 Datasets

Among the datasets listed below, we transformed those with multi-class classification tasks into multiple binary classification tasks. This augmentation resulted into 666 datasets. We

**Algorithm 1:** LinUCB exploration policy for sequential AutoML

**Input:** Exploration budget $c$, knowledge $R(0)$ (empty, i.e. 0 rows), $\alpha \in R^+$, $s$
length of burn-in phase and number of steps between updates

**1** $A(0)_k \leftarrow I_K$ (identity) and $b(0)_k \leftarrow 0_{K,1}$ for each pipeline $k = 1...K$ ;

**2** Set $\tilde{P}(0) \in \mathbb{R}^{L \times K}$ at random ;

**3 for** *each episode* $t = 1, \ldots, T$ **do**

**4**    **if** $t \leq s$ *(burn-in phase)* **then**

**5**       $\mathcal{C}(t) \leftarrow$ select pipelines uniformly at random

**6**    **end**

**7**    **else**

**8**       **for** *each pipeline* $k = 1, \ldots, K$ **do**

**9**          $\theta_k \leftarrow A(t-1)_k^{-1} b(t-1)_k$;

**10**          $\text{UCB}(t)_k \leftarrow \langle \tilde{P}_{\cdot,k}(t-1), \theta_k(t) \rangle + \alpha \sqrt{\tilde{P}_{\cdot,k}(t-1)^\top A(t-1)_k^{-1} \tilde{P}_{\cdot,k}(t-1)}$;

**11**       **end**

**12**       $\mathcal{C}(t) \leftarrow$ select the top-$c$ pipelines with the largest $\text{UCB}(t)_k$ ;

**13**    **end**

**14**    $R'(t) \leftarrow R(t-1)$ with added $t$-th row ;

**15**    **for** *each pipeline* $k \in \mathcal{C}(t)$ **do**

**16**       Observe $r_{t,k}$ and store it in $R'(t)$ ;

**17**    **end**

**18**    $k_t, \leftarrow$ select using recommendation policy;

**19**    **if** $k_t \notin C(t)$ **then**

**20**       $R(t) \leftarrow$ update $R'(t)$ with $r_{t,k_t}$ ;

**21**    **end**

**22**    **else**

**23**       $R(t) \leftarrow R'(t)$

**24**    **end**

**25**    **if** $t \mod s = 0$ *(update step)* **then**

**26**       $\tilde{P}(t) \leftarrow$ matrix factorization of $R(t)$

**27**    **end**

**28**    **else**

**29**       $\tilde{P}(t) \leftarrow \tilde{P}(t-1)$

**30**    **end**

**31**    **for** $k \in C(t) \cup \{k_t\}$ **do**

**32**       $A(t)_k \leftarrow A(t-1)_k + \tilde{P}_{\cdot,k}(t)\tilde{P}_{\cdot,k}(t)^\top$ ;

**33**       $b(t)_k \leftarrow b(t-1)_k + r_{t,k}\tilde{P}_{\cdot,k}(t)$ ;

**34**    **end**

**35 end**

keep 140 datasets apart for the KNN exploration policy. In the end, we use a sequence of 526 datasets for these experiments.

List of datasets selected from UCR repository (Dau et al., 2019): ACSF1, Adiac, Arrow-Head, BME, Beef, BeetleFly, BirdChicken.csv, CBF, Car, Chinatown.csv, ChlorineConcentration, CinCECGTorso, Coffee, Computers, CricketX, CricketY, CricketZ, Crop, DiatomSizeReduction, DistalPhalanxOutlineAgeGroup, DistalPhalanxTW, ECG200, ECG5000, ECGFiveDays, EOGHorizontalSignal, EOGVerticalSignal, Earthquakes, FaceAll, FaceFour, FacesUCR, FiftyWords, Fish, FreezerRegularTrain, FreezerSmallTrain, Fungi', GunPoint, GunPointAgeSpan, GunPointMaleVersusFemale, GunPointOldVersusYoung, Ham, Haptics, Herring, HouseTwenty, InlineSkate, InsectEPGRegularTrain, InsectEPGSmallTrain, InsectWingbeatSound, ItalyPowerDemand, LargeKitchenAppliances, Lightning2, Lightning7, Mallat, Meat, MedicalImages, MiddlePhalanxOutlineAgeGroup, MiddlePhalanxTW, NonInvasiveFetalECGThorax1, NonInvasiveFetalECGThorax2, OSULeaf, OliveOil, Phoneme, PigAirwayPressure, PigArtPressure, PigCVP, Plane, PowerCons.csv, ProximalPhalanxOutlineAgeGroup, ProximalPhalanxTW, RefrigerationDevices, ScreenType, SemgHandGenderCh2, SemgHandMovementCh2, SemgHandSubjectCh2, ShapeletSim, ShapesAll, SmallKitchenAppliances, SmoothSubspace, SonyAIBORobotSurface1, SonyAIBORobotSurface2, Strawberry, SwedishLeaf, Symbols, SyntheticControl, ToeSegmentation1, ToeSegmentati Trace,TwoLeadECG, UMD, UWaveGestureLibra Wine, WordSynonyms, Worms, WormsTwoClass

## B.2 Pipelines

In order to obtain a ground truth required for computing the cumulative regret (Equation 1), we evaluated the PRAUC performance of each pipeline on each dataset using a 5-folds cross-validation. A exhaustive list of the pipelines included is available in Table 1.

## B.3 Description of the obtained benchmark

The descriptive statistics in Figure 2 are obtained from the exhaustive benchmark of the 175 pipelines (see Section B.2) over the 666 datasets (see Section B.1).

The descriptive statistics in Figure 2 are obtained from the exhaustive benchmarking. The best pipelines are most often obtained by applying a weighted loss to mitigate the imbalance ratio of the dataset (sub-figure 1). Indeed, the minority class represents less than 0.1 of the observations for more than 50% of the datasets (sub-figure 8), which corresponds to an important imbalance ratio. Overall, the datasets included in the benchmarking have heterogeneous characteristics such as the number of observations, or the number of features (sub-figures 6-7). More than half of the datasets find an optimal pipeline with a performance higher than 0.85 in PRAUC suggesting that the 175 available pipelines cover well the variety of datasets (sub-figure 2). There exist groups of pipelines with equivalent performances. The smallest regret (sub-figure 4) is equal to 0 for more than 50% of the datasets suggesting that most of the time the recommendation policy can identify at least 2 well performing pipelines close to the optimum, which makes the pipeline optimization task easier. However, there are important performance amplitudes between the 175 available pipelines as the largest regret is larger or equal to 0.6 in 50% of the datasets (sub-figure 3). The performance map (sub-figure 5) suggests the existence of clusters of pipelines with equivalent performances.

| Component | Algorithm | Hyper-parameters |
|---|---|---|
| Imbalance Management | Random Over Sampling | the 2 labels are sampled equally (50% each) |
| | Random Under Sampling | the 2 labels are sampled equally (50% each) |
| | Weighted Loss | proportion of each label in the training set |
| | None | None |
| Classifier | Gradient Boosting | n_estimators(100)
learning_rate(0.1)
subsample(0.5, 0.7, 0.9)
max_depth(3, 4, 5) |
| | Support Vector Machine | C(0.01, 1, 10, 1000)
kernel(linear, rbf) |
| | K-Nearest Neighbor | leaf_size(3,10,50,100) |
| | Radius Neighbor | radius(0.8, 1, 1.2)
leaf_size(10, 30, 50)
outlier_label('most_frequent') |
| | Random Forest | n_estimator(100)
max_depth(3, 5, 10, 50, 100) |
| | Logistic Regression | regularization(L1)
alpha(0.0001, 0.01, 0.1, 0.5, 0.8) |
| | Multi Layers Perceptron | layer1(100)
layer2(50) |

Table 1: Pipeline Decision Space

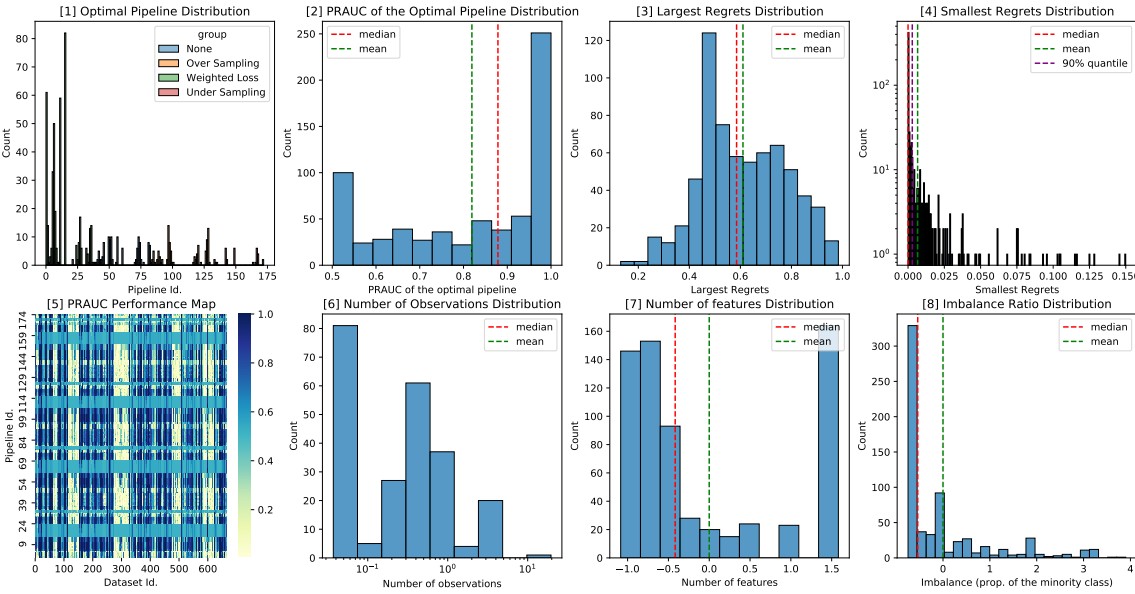

Figure 2: Descriptive Statistics obtained from the exhaustive benchmarking of 175 pipelines over the 666 datasets. [1] counts how many time each of the 175 pipeline is the optimal one, [2] shows the distribution of the optimal PRAUC performance, [3] shows to the distribution of the largest regrets, [4] shows to the distribution of the smallest regrets, [5] is a map of the PRAUC performance over all the datasets and pipelines, [6] depicts the number of observations per dataset, [7] depicts the number of features per dataset, [8] shows the distribution of the imbalance ratio (proportion of the minority class).

Indeed, some pipelines are prone to perform equally on the same dataset because they use the same model and are not sensitive to resampling operations (for instance, KNN); or because the set of hyper-parameters between 2 pipelines using the same model is very close.

## Appendix C. Parameter setting used in the Experiments

### C.1 LinUCB

Recall that the burn-in phase and the update step (see Section 4) size are controlled by the same parameter $s$. We considered different numbers of steps between updates $s \in \{10, 100\}$. In Figure 1 (Section 5.3) we show the results for $s = 10$ since it led to better performance compared to the case $s = 100$ showcased in the Figure 3. We consider different values of optimism $\alpha \in \{0.1, 0.01\}$.

### C.2 Matrix factorization latent representations

For each matrix factorization algorithm, we set the number of latent dimensions to $L = 40$. The latent matrices $\tilde{Q}(0)$ and $\tilde{P}(0)$ are initialized at random. As for the regularization

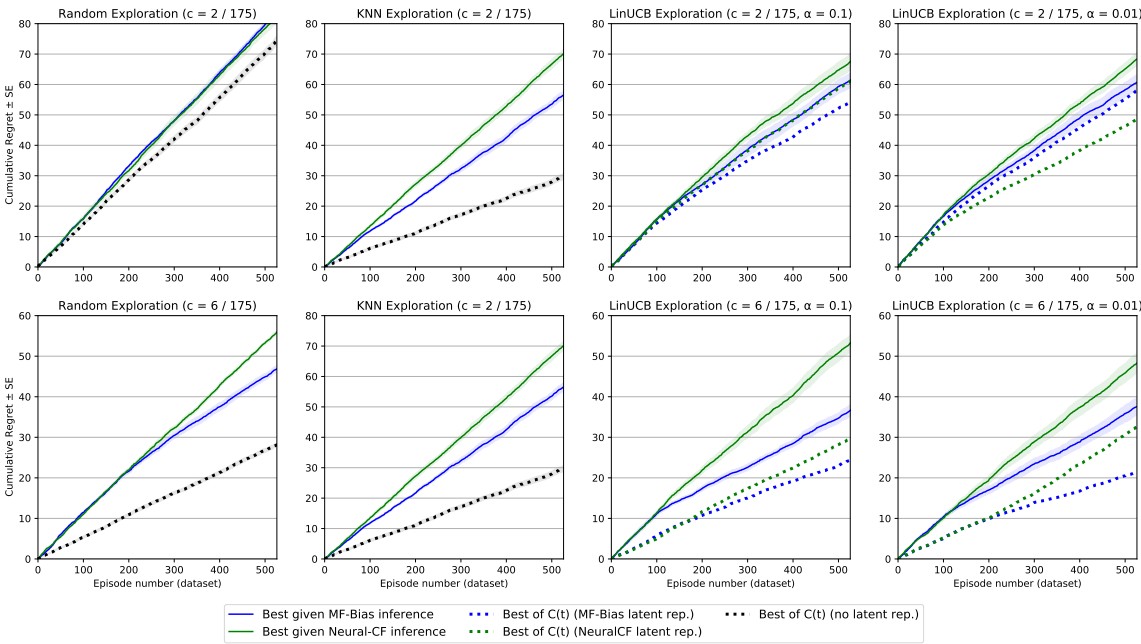

Figure 3: Cumulative regret averaged over a 10-folds cross validation with the resulting standard error using $s = 100$ steps between matrix factorization updates. Lower is better.

parameters, a grid search including values ranging from 0.1 to 0.00001 lead us to choose $\lambda_Q = \lambda_P = 0.01$ because it shown satisfying results. The gradient of each weight is individually clipped so that its norm is not higher than 1 (the default value on Tensorflow 2.0). The learning rate of the stochastic gradient descent is set to the default value on Tensorflow 2.0 $\gamma = 0.01$. Each algorithm uses 75 epochs for the training.

**C.3 Meta-features used in the KNN exploration policy**

The list of meta features used in the KNN exploration policy is: Size of the file (Mb), Number of classes (2), Number of observations (time-series), log (Number of observations), Number of features(timesteps), log (Number of features), ratio Number of Obsevations / Number of Features, Entropy of the label, Skewness (Mean, Standard-deviation, Minimum, Maximum), Kurtosis (Mean, Standard-deviation, Minimum, Maximum), Standard Deviation (Mean, Standard-deviation, Minimum, Maximum), Variation Coefficient (Mean, Standard-deviation, Minimum, Maximum).

