# OpenReview forum: "Sequential Automated Machine Learning: Bandits-driven Exploration using a Collaborative Filtering Representation"
_ICML.cc/2021/Workshop/AutoML — AutoML@ICML2021 Poster_

### Official Review · Reviewer_273s · 2021-06-05

**Rating:** 7
**Confidence:** 3

**Review:**

The authors of this paper propose an algorithm called LinUCB to make recommendations for a pipeline (preprocessing, algorithm, hyperparameters) for a sequence of datasets.

- In general the paper is well written, although it is not easy to understand for scientist without prior knowledge in collaborative filtering and matrix factorization.
- The part 4.2. might be understandable for people with knowledge in matrix factorization but is not easily understandable by an average data scientist.
- The algorithm seems to be useful compared to other strategies in their paper. It is not completely clear why they choosed the knn strategy for comparison although this might be one of the "standard" strategies in the literature (currently).
- The random strategy seems to makes sense as a default strategy to compare to.
- x and y labels in figure 1, 2 and 3 are quite hard to read especially for people with non-perfect eyes.
- I could not find any big orthographical mistakes, just small mistakes such as using "an sequential setting" or a missing "." in "e.g.".
- It is not clear why for some papers links are provided in the "References" section and for others not.

---

### Official Review · Reviewer_vBRu · 2021-06-10
**Promising Idea with Limited Results**

**Rating:** 6
**Confidence:** 3

**Review:**

The authors combine two collaborative filtering recommendation algorithms (Neural CF and MF-Bias) with bandits-driven exploration methods.

The paper starts promisingly: It is mostly well written, the problem is well motivated and the proposed approach is relatively clearly described.  The clarity, though, could be improved in some parts. For instance, in section 2, the authors define "K" as the "available" pipelines but what does "available" mean? The total number of theoretically possible pipelines? Or the number of pipelines for which (for other datasets) performance measures are available? Later, this becomes clear, but I suggest the authors clarify this already in section 2.

My major criticism relates to the result section and the lack of baselines. The main result (figure 1) shows 8 charts, with the episode number on the x-axis and regret on the y-axis. The charts show the results of 5 algorithms/variations, each proposed by the authors. There is no baseline, e.g. a normal AutoML tool or some other state-of-the-art approach (e.g. Cunha 2018). As such, the contribution of this manuscript is very limited. Similarly, the cumulative regret as the final metric is, in my opinion, not ideal. Much more interesting would "accuracy", i.e. how often the actual best pipeline(s) out of the 175 were selected.

It would also be nice if the run-time and memory requirements would be discussed (also in comparison with other baselines).

Last, but not least, the related work section could be more comprehensive. Especially more work on recommendation algorithms being used for algorithm selection should be discussed https://scholar.google.at/scholar?hl=en&as_sdt=0%2C5&q=recommendation+algorithm+selection&btnG=

Overall, despite my criticism, I recommend a borderline / just accept, because I believe the paper could stimulate some good discussions.

Minor Issues:

The authors title their manuscript with *Sequential* AutoML where datasets arrive one after another. I fail to see the sequential nature of the work. To me, it appears that each of the authors' approaches selects the pipelines 'c' regardless of each other, and not in a sequence. If this is right, the authors should adjust the wording of their paper. If I misunderstood something, maybe the authors could explain it in more detail in the manuscript?

The paper contains a link to the author's GitHub repository, which compromises the anonymity of the author.

---

### Official Review · Reviewer_iukr · 2021-06-13
**The motivation needs to be clarified further**

**Rating:** 5
**Confidence:** 4

**Review:**

This paper considers an online AutoML task, where the learner is receiving ML tasks sequentially and needs to recommend a high-performing pipeline, among a pre-defined set of pipelines, to the newly-arrived task (dataset).

The authors propose to cast the problem as a recommender system where the pipelines corresponds to the items and datasets to the users. At each time, the learner can choose a small subset of pipelines from the entire pipeline pool to test on the new dataset before recommending a final pipeline. The authors then propose to use LinUCB to deal with the exploration part by leveraging the information inferred from a MF constructed based on the previous observations.

The paper is mostly well written. The idea is clearly explained and I think it makes sense given the problem formulation. Personally, there are not many novelties in the proposed method, as LinUCB is well known to be a good candidate for recommender systems and CF techniques are also well developed. I am ok with that as it is also a good contribution to thoroughly test existing methods to a new problem (as is done in the paper, the experiments are quite exhaustive). My major concern about the paper, however, is about the problem setting itself. Essentially, it is somewhat difficult for me to see in what kind of scenario would this online setting be realistic. Usually a company would not have thousands of hundred of ML tasks to tackle everyday.  Another point that annoys me a bit is the manually-chosen parameter c. For some very large ML models, even only train a few of them on every new dataset would be unaffordable. It is probably interesting to propose a self-tuning process on c.

In summary, this could be an ok paper for the workshop, but personally I think at least the motivation needs to be further discussed.

---

### Decision · Program_Chairs · 2021-06-21

Accept (Poster)